# Stating Failure Modelling Limitations of High Strength Sheets: Implications to Sheet Metal Forming

**DOI:** 10.3390/ma14247821

**Published:** 2021-12-17

**Authors:** Olle Sandin, Pär Jonsén, David Frómeta, Daniel Casellas

**Affiliations:** 1Division of Solid Mechanics, Luleå University of Technology, SE-971 87 Luleå, Sweden; par.jonsen@ltu.se (P.J.); daniel.casellas@eurecat.org (D.C.); 2Eurecat, Centre Tecnològic de Catalunya, Unit of Metallic and Ceramic Materials, Plaça de la Ciència 2, 08243 Manresa, Spain; david.frometa@eurecat.org

**Keywords:** advanced high strength steel, GISSMO, complex phase steel, failure modelling

## Abstract

This article discusses the fracture modelling accuracy of strain-driven ductile fracture models when introducing damage of high strength sheet steel. Numerical modelling of well-known fracture mechanical tests was conducted using a failure and damage model to control damage and fracture evolution. A thorough validation of the simulation results was conducted against results from laboratory testing. Such validations show that the damage and failure model is suited for modelling of material failure and fracture evolution of specimens without damage. However, pre-damaged specimens show less correlation as the damage and failure model over-predicts the displacement at crack initiation with an average of 28%. Consequently, the results in this article show the need for an extension of the damage and failure model that accounts for the fracture mechanisms at the crack tip. Such extension would aid in the improvement of fracture mechanical testing procedures and the modelling of high strength sheet metal manufacturing, as several sheet manufacturing processes are defined by material fracture.

## 1. Introduction

Over the past decades, the automotive industry has continuously strived to reduce the product development time in order to increase their competitiveness and profit. By the use of numerical modelling with the Finite Element method (FEM), it is now possible to perform virtual testing of components with high accuracy, thus reducing the amount of costly experiments and prototyping. However, with extended use of computer aided engineering in the product development process comes new challenges and demands on the numerical FE-modelling. One of the challenges of numerical FE-modelling is how to accurately describe damage generation in terms of micro-cracks, sharp notches and voids. Damage markedly affects sheet formability and part performance of high strength metallic sheets, such as Advanced High Strength Steels (AHSS) or high strength aluminium alloys. Most manufacturing defects in such materials are related to the edge cracking phenomenon, that is, the generation of small cracks at sheared (damaged) edges during stretching or bending. Regarding part performance, crash resistance involves the generalized cracking (damage) prior to final fracture. High strength metallic sheets are widely used for automotive lightweight constructions. Thus, FE-analysis of sheet metal forming and crash modelling requires numerical models that accurately handle damage, that is, the formation and propagation of cracks.

Shear cutting of high strength sheet metals is such process where the material separation involves the formation of cracks, and inherited damage from the cut affects the formability of the part. Publications by Konieczny and Henderson [1], Dykeman et al. [2], Thomas [3] and Sigvant et al. [4] show that the shear cutting process degrades the stretch-flangeability of the material when comparing formability of sheared edges to laser-cut or polished edges. This statement is further supported through work by Mori et al. [5], Mori et al. [6], Gläsner et al. [7] and Saengkhiao et al. [8], who present warming and smoothing techniques that improve the cut edge quality, thus improving the sheet formability. Microscopical investigations of sheared cut edges performed by Wu et al. [9] and Yoon et al. [10] also show the formation of micro-cracks along the cut edge surface, which are considered to mainly cause the limited formability of the cut edge due to edge-cracking.

Casellas et al. [11] showed that the Essential Work of Fracture (EWF), a mechanical property determined by fracture mechanics that is equivalent to the conventional Elastic Plastic Fracture Mechanics (EPFM) toughness value $J_{c}$ and measures crack propagation resistance, effectively describes stretch-flangeability and edge cracking resistance of AHSS. Such work pointed out that crack-related phenomena in sheet metal forming can be addressed by fracture mechanics concepts. Frómeta et al. [12] proposed to use the EWF to rank crashworthiness in AHSS, because in crash tests the generation of damage (cracks) and the further crack propagation is related to the overall crash energy absorption. The work shows excellent correlation between crack propagation resistance, that is, fracture toughness, and crashworthiness for different AHSS grades. Recently, Frómeta et al. [13] proposed to use fracture mechanics parameters related to the EWF to understand cracking resistance of AHSS and describe the damage that limits the formability of AHSS. The above publications related to the fracture toughness and cracking resistance of AHSS suggest that damage during forming and part performance should be addressed considering the crack nucleation and propagation. This directly implies that numerical modelling of shear cutting and forming of AHSS also should be capable to capture and follow the crack nucleation and propagation.

Numerical modelling of the shear cutting process is a challenging forming operation to simulate, as shear cutting involves both large material deformation, shear failure and duplex crack initiation. In addition, the numerical failure model should be able to handle a large range of stress states. The complex failure assessment is conventionally managed by strain-driven ductile failure models with stress state dependency. This was done by Hambli and Potiron [14], who implemented a Lemaitre damage model presented by Lemaitre and Chaboche [15] for an axisymmetrical shear cutting model of 1060 steel for process optimisation purposes. Hambli and Potiron [14] defined the crack initiation and propagation through loss of element stiffness when a critical damage value was obtained in the element. Similarly, Thipprakmas et al. [16] utilised an axisymmetrical model along with a Rice and Tracey [17] fracture criterion to analyse fine-blanking of medium steel S45C. In order to avoid numerical divergence due to distorted elements, a re-meshing algorithm adjusted the elements of the shear affected zone. To ensure a correct crack path, Bacha et al. [18] developed a two-step model for trimming of aluminium consisting of re-meshing and arbitrary Lagrangian formulation of the shearing process until crack initiation, from where the plastic equivalent strain acted as a damage variable for the following crack propagation. Work that has received considerable attention was presented by Wang et al. [19] and consisted of a three dimensional punching and forming model where the modified Mohr–Coulomb model by Bai and Wierzbicki [20,21] defined the material failure. Wang et al. [19] could predict edge cracking of high strength steel during hole expansion with inherent edge damage from shear cutting, using element deletion to simulate material failure. The works mentioned have in common that they utilise strain-driven ductile failure models or limiting failure strain values for simulating the fracture process through element deletion or stiffness reduction. The failure strain values are mainly determined from experimental tensile testing. This makes them suitable for the global failure analysis of undamaged materials, such as for damage initiation in crash analysis. Applying strain-driven ductile failure models in crack initiation analysis will, according to Anderson [22], cause mesh size dependency and often need fine tuning to fit the experimental data. This procedure contrasts with fracture mechanical methodologies, such as the J-integral, where energy measures define the crack opening and propagation. Fracture mechanical approaches generally focus on the analysis of crack tip conditions and are nearly independent of the size of the crack tip. However, as stated by Anderson [22], incorporating fracture mechanical approaches for FE-frameworks often requires cumbersome re-meshing, which makes fracture mechanical techniques seldom used in commercial fracture analysis.

It is clear that the intended area of usage differs between strain-driven ductile failure models and fracture mechanics, but for the numerical analysis of the shear cutting process there is an apparent need for both a failure assessment of undamaged material and crack tip analysis. To investigate the applicability of using only strain-driven ductile failure models for processes involving cracks, such as shear cutting, the authors of this article perform numerical modelling of cracked high strength sheet steel without taking notch effects or crack-tip singularities into account. The current work focuses on well-known fracture mechanical lab tests in order to ensure controlled crack initiation and propagation. Results from the laboratory tests were compared with corresponding simulation results using the Generalised Incremental Stress State Dependent Damage Model (GISSMO) as a ductile failure model in order to investigate how cracks limit the modelling accuracy. The laboratory tests experience Mode I crack opening with a minimum of uncertain details such as friction, dynamics or joints. The simplicity of the validation cases makes them suitable when evaluating the numerical modelling as every assumption made for the model can cause deviation from the actual load case. The material investigated is a complex phase steel with an ultimate tensile strength of 1000 MPa approximately and high ductility (named as CP1000HD from now). It is an AHSS grade with 1.5 mm thickness, commonly used for automotive crash components due to its high strain hardening rate and ductility.

## 2. Introduction to Damage- and Failure Modelling

In this work, the Generalised Incremental Stress State Dependent Damage Model (GISSMO) was used for damage and failure modelling. GISSMO was developed by Neukamm et al. [23] and extended to include lode angle dependency by Basaran et al. [24]. Further description of the GISSMO model was presented by Andrade et al. [25]. GISSMO was implemented for the FE-software LS-Dyna and is a phenomenological damage and failure model that uses stress state dependent equivalent plastic strain as failure criteria. It is uncoupled from the plasticity model, thus compatible with several of the available material models in the LS-Dyna library. Ever since the implementation of GISSMO in LS-Dyna has its ability to generate high accuracy results been on display in several publications. Omer et al. [26] used GISSMO for the modelling of crush members and Chen et al. [27] evaluated the fracture predictability of an AHSS GISSMO model through a number of validation crash simulation. Similarly, Pérez Caro et al. [28] could detect forming damages of Alloy 718 through the use of GISSMO. Considering its well documented ability to model failure, it is made clear that GISSMO is a useful tool for detecting where and when material failure occurs for certain purposes.

The damage accumulation rule is defined by Neukamm et al. [23] according to Equation (Equation 1), where $\dot{D}$ denotes the incremental damage value, $\varepsilon_{f}\left(\eta \right)$ is the stress state dependent failure strain value, ${\dot{\varepsilon }}^{p}$ is the accumulated plastic strain rate and *n* is the damage exponent.
(1)$$\dot{D}=\frac{n}{\varepsilon_{f}\left(\eta \right)}D^{\left(1-\frac{1}{n}\right)}{\dot{\varepsilon }}^{p}.$$

When the accumulated damage value reaches D=1 material failure occurs. If wanted by the user, material failure can be preceded by material degradation through coupling stresses to the damage level. The damage/stress coupling is displayed in Equation (Equation 2), where it is shown how the stress is coupled through the accumulated damage value *D* and the damage value at material instability $D_{crit}$. The fading exponent *m* controls the non-linearity and the exponential damage coupling.
(2)$$\sigma^{*}=\sigma \left(1-{\left(\frac{D-D_{crit}}{1-D_{crit}}\right)}^{m}\right).$$

The material instability value *F* controls when damage/stress coupling is initiated and accumulates in a similar way as the damage value *D*, as shown in Equation (Equation 3). An accumulated instability value of F=1 defines the point of diffuse necking and the damage/stress coupling is initiated.
(3)$$\dot{F}=\frac{n}{\varepsilon_{i}\left(\eta \right)}F^{\left(1-\frac{1}{n}\right)}{\dot{\varepsilon }}^{p}.$$

The instability strain $\varepsilon_{i}\left(\eta \right)$ can either be a fixed value or a stress state dependent curve. Material instability or point of diffuse necking is a complex phenomenon and several instability criteria have been developed over the years. A commonly used instability criteria worth mentioning is the Considére criterion, defined as the strain at maximum engineering stress with a corresponding plastic strain equal to the work hardening exponent, presented by Considére [29]. The damage and failure implementation provides the option to use arbitrary curves and functions to describe the material instability, denoted *ECRIT*. Consequently, it is up to the user to determine how to define the material instability criteria that suit the particular material of interest.

Not seldom is the stress state dependent failure strain and material instability obtained through parameter calibration, where parameters such as $\varepsilon_{f}\left(\eta \right)$, *m*, *n* and $\varepsilon_{i}\left(\eta \right)$ are fitted to give a correlation between simulation and corresponding test results.

Material failure in finite element analysis is generally manifested through element erosion when the failure strain $\varepsilon_{f}\left(\eta \right)$ is obtained for the element. However, the actual effective strain value of an element is affected by the element length, especially for geometries where stress/strain localization occurs. The element length dependency of the effective strain gives a mesh dependency during failure modelling that needs to be accounted for. The mesh dependency is treated through mesh regularisation, where a factor defined by the user scales the failure strain value. The user can input a curve defining scale factor versus element size in order to cover the mesh dependency over a range of element lengths.

The recent development of damage and failure modelling includes anisotropic hardening and failure behaviour. This development is driven by the urge of predicting failure in complex geometries consisting of anisotropic materials such as high strength steel, aluminium and composites. Such work was presented by Park et al. [30], which modified the Lou–Huh ductile fracture criterion based on the Hill’s 48 anisotropic yield function, thus incorporating the anisotropic effect of the material in the failure strain. Furthermore, Basak et al. [31] showed that the formability and failure of deep-drawn anisotropic materials can be predicted using anisotropic material models along with MK-FLD and GISSMO. The extension of GISSMO presented by Koch et al. [32] also meant a damage and failure model with similar capabilities as its precursor, but with the possibility of including individual damage and failure characteristics depending on the deformation direction. Anisotropic behaviour of the CP1000HD was not considered in this work because it is expected to have a low impact on the modelling results. First, the specimens are thin (1.5 mm) and, secondly, the crack imitation and propagation occur locally under pure tensile loading. In complex tensile states situations, such as punching or shear, anisotropy effects should be considered.

A detailed description of the damage and failure model and element erosion techniques for LS-Dyna is found in the LS-Dyna user’s manual by the Livermore Software Technology [33] and the theory manual by Hallquist [34].

## 3. Materials and Methods

### 3.1. Material Characterisation

The material behaviour at different stress states (or stress triaxiality) was investigated by tensile testing of four different notched specimen geometries (shown in Figure 1). The tensile specimen geometries for trixiality testing were presented by Sjöberg et al. [35]. These tests were performed perpendicular to the sheet rolling direction and enabled the calibration of the CP1000HD fracture resistance and fracture locus. A Stiebler plastic hardening model by Stiebler et al. [36] was calibrated through inverse modelling in order to describe the plastic strain true stress characteristics of the CP1000HD material. The inverse modelling scheme performed parameter calibration of the plastic hardening material constants *A* to *D* and the Young’s modulus *E* such that an FE-model of the R3.75 tensile test reproduced the force-elongation results of the corresponding test. The FE-model is presented in detail in Section 3.4, and Figure 2a) shows the calibration procedure flowchart. During calibration of the plastic hardening model, there was material damage and failure disabled in the FE-model. The Steibler plastic hardening model is presented in Equation (Equation 4) and the calibrated parameters *A* to *D* of the Stiebler model are stated in Table 1.
(4)$$\sigma \left(\varepsilon_{p}\right)=A+B\varepsilon_{p}+C\left(1-\exp\left(-D\varepsilon_{p}\right)\right).$$

The true plastic strain–true stress curve of the CP1000HD material is shown in Figure 3. The plastic hardening curve shown in Figure 3 gives the plastic hardening behaviour of the material as tabulated data to a von Mises isotropic piecewise linear plasticity model. Table 2 presents the tensile properties at different rolling directions for the investigated CP1000HD.

Digital image correlation (DIC) and strain field evaluation provided the possibility of measuring the local strain field of the necked specimen at crack initiation. The Stepwise Modelling Method developed by Marth et al. [37] enabled the calculation of equivalent plastic strain and stress triaxiality at failure. The SMM extracts the stress tensor from an integration path that forms along the localised zone of the tensile specimen, consisting of several facets. Within each facet is a stress tensor $\sigma_{ij}$ calculated from which it is possible to determine the mean stress $\sigma_{m}$ and the effective von Mises stress ${\bar{\sigma }}_{vM}$. The mean and effective stresses form the stress triaxiality η according to Equation (Equation 5). Sjöberg et al. [35] provide a detailed description of the stress trixiality calculation in SMM. The facet size during strain field evaluation was set to 0.1 mm with 75 % facet overlap in order to obtain a high resolution strain field. All samples were prepared with the rolling direction of the sheet metal perpendicular to the pulling direction of the tensile test.
(5)$$\eta =\frac{\sigma_{m}}{{\bar{\sigma }}_{vM}}.$$

With the stress-triaxiality and failure strain data from the four tensile testing specimen geometries in Figure 1, it was possible to calibrate a modified Mohr–Coulomb (MMC) fracture locus for plane stress according to the work of Bai and Wierzbicki [20,21]. The MMC fracture locus is described by Equation (Equation 6), where parameters $C_{1}$ to $C_{5}$ constitute the curve shape. A least-square curve fit enabled the calibration of parameters $C_{1}$ to $C_{5}$ such that the MMC plane stress curve fitted the experimental triaxiality and failure strain values. The impact of each parameter $C_{1}$ to $C_{5}$ is described in detail by Bai and Wierzbicki [21] and the values of $C_{1}$ to $C_{5}$ for the CP1000HD material are shown in Table 3. The function variables η and $\bar{\theta }$ denote stress triaxiality and lode angle parameters, respectively.
(6)$$\begin{array}{cc} \; & \varepsilon_{f}\left(\eta ,\bar{\theta }\right)=\{C_{2}\left[C_{3}+\frac{\sqrt{3}}{2-\sqrt{3}}\left(C^{ax}-C_{3}\right)\left(sec\left(\frac{\bar{\theta }\pi }{6}\right)-1\right)\right]\cdot \\ \; & \cdot \left[\sqrt{\frac{1+C_{1}^{2}}{3}}cos\left(\frac{\bar{\theta }\pi }{6}\right)+C_{1}\left(\eta +\frac{1}{3}sin\left(\frac{\bar{\theta }\pi }{6}\right)\right)\right]{\}}^{-\frac{1}{C_{5}}}, \end{array}$$
where
(7)$$C^{ax}=\left\{\begin{array}{cc} 1 & \mathrm{for}\bar{\theta }\geq 0\\ C_{4} & \mathrm{for}\bar{\theta }<0. \end{array}\right.$$

As plane stress conditions are assumed for sheet metal, the lode angle parameter in Equation (Equation 6) can be expressed as a function of the stress triaxiality according to Bai and Wierzbicki [20,21], as shown in Equation (Equation 8).
(8)$$\bar{\theta }=1-\frac{2}{\pi }cos^{-1}\left(-\frac{27}{2}\eta \left(\eta^{2}-\frac{1}{3}\right)\right).$$

The calibrated MMC curve for plane stress is shown in Figure 4. Figure 4 also shows the strain versus triaxiality paths until failure for each test, as well as the strain values at maximum force. Triaxiality values obtained in the in-plane shear testing are negative, probably due to the experimental difficulties associated with the machining and testing of shear specimens (Figure 1d). It might lead to the inaccurate prediction of shear failure, but would have no influence when failure occurs under tensile stress states.

For solid elements, it is necessary to include lode angle parameter dependency for the fracture locus, thus obtaining a surface of failure strain values that spans the full ranges of triaxialities and lode parameter values. With the the MMC parameters $C_{1}$ to $C_{5}$, known for the plane stress assumption, it was possible to construct an MMC failure surface through input of −1/3≤η≤2/3 and $-1\leq \bar{\theta }\leq 1$ in Equation (Equation 6). The MMC surface is shown in Figure 5, where the plane stress MMC curve is shown in red.

### 3.2. CP1000HD Damage and Failure Model

The experimental triaxiality testing data presented in Section 3.1 were used as initial input data when calibrating the CP1000HD damage and failure model. The MMC failure surface, consisting of the triaxiality, failure strain and lode angle parameter relationship presented in Section 3.1, was input to the damage and failure model to define stress state dependent element erosion values. The MMC surface was discretised with 20 lode angle steps and 50 triaxiality-failure strain steps.

Through the use of the optimisation software LS-OPT, the failure strain-trixiality curves were scaled in order for the numerical modelling results from the R3.75 FE-model to better match the force displacement tensile testing response of the R3.75 specimen. The fading exponent *m* and damage exponent *n* were also calibrated along with the failure strain-trixiality curves. The damage and failure model calibration flowchart is shown in Figure 2b). Figure 6 displays the final MMC surface after calibration along with the calibrated values of *m* and *n* in Table 4. More insight into the parameter calibration using LS-OPT is found in the LS-OPT user’s manual by Stander et al. [38].

The characteristic element length of the R3.75 specimen mesh used throughout the damage and failure model calibration was 0.1 mm. No mesh regularisation was performed as the damage and failure model model was intended to use on models with the same element size. The authors of this article defined the instability point for this particular CP1000HD material as the $\varepsilon_{i}\left(\eta \right)$ when maximum force of the tensile test is reached and the experimental instability points constituted a second order instability curve to define the *ECRIT* data for the damage and failure model model.

### 3.3. Experimental Validation

Validation of the numerical models was done by experimental testing of the standardised VDA 238-100 bending test and by double edge notched tensile (DENT) testing. In both tests, the cracks initiate and propagate under Mode I, that is, constantly under tensile loading. The bending test specimens have been machined and do not present cracks or surface defects. Meanwhile, two types of specimens were prepared for DENT; with and without fatigue pre-cracks, that is, with and without damage. The presence of the fatigue cracks in the DENT specimen provided the opportunity to investigate the effect of crack tip stress concentration and compare the results to uncracked (undamaged) material loaded in bending and tension. The isotropic piecewise plasticity model presented in Section 3.1 was calibrated against experimental results obtained from tensile testing performed perpendicular to the sheet rolling direction. The VDA 238-100 and DENT testing were consequently performed with sheet placement perpendicular to the rolling direction as well. Anisotropic behaviour of the CP1000HD material was not examined as the crack initiation during the specific experiment geometries in this article are not affected by anisotropy. The crack initiation and opening occurs locally under pure tensile loading. For complex geometries, anisotropic effects should be considered.

#### 3.3.1. VDA 238-100

The VDA 238-100 bending test widely used in the automotive industry to characterise the local formability of metal sheets. It is described with the VDA 238-100 [39] standard by Verband der Automobilindustrie (VDA). It is widely used in the automotive industry, for instance when determining the foldability of crash components. A schematic image of the experimental setup is shown in Figure 7. The test consists of a sheet metal specimen, a bending punch with 0.4 mm edge radius and two support rollers. The sheet metal sample of 60 × 60 mm is placed on the rollers and it is being bent by a downward stroke of the bending punch. The VDA 238-100 standard specifies that the bending punch stroke is ended when the force drops 60 N below the peak force. After the test is ended the deformed sheet metal specimen is examined for cracks and the bending angle is measured. Crack detection from VDA 238-100 bending tests is done visually or by optical measurement.

Measurements of the elastic deformation of the tool accounted for the vertical stiffness of the bending punch. One exception from the VDA 238-100 recommendation was made during the testing presented in this article. Instead of using rollers mounted on bearing, the rollers were fixed and lubricated by teflon paper. The modification of the experimental setup was not considered to influence the validity of the results.

#### 3.3.2. Double Edge Notched Tensile (DENT) Tests

The double edge notched tensile (DENT) test is increasingly used for measuring fracture toughness of thin sheets according to Casellas et al. and Frómeta et al. [11,12,13]. The configuration of the DENT specimens ensures pure Mode I crack propagation during tensile tests. Two types of specimens were prepared for DENT; (I) notched specimens with a notch of 150 micron radii, machined by electro discharge machining; and (II) pre-cracked specimens, with fatigue pre-cracks generated at the notch root generated by fatigue tests, following the procedure described by Frometa et al. [40]. Figure 8 schematically shows the notched and cracked DENT specimen geometry. Table 5 shows the tested ligament length for notched and cracked DENT specimens.

Results from notched and cracked DENT specimens are used to evaluate how accurately the ductile failure model can mimic the failure behaviour of AHSS thin sheets with and without the presence of cracks. In notched DENT specimen, the notch root endures substantial plastic work and necking before fracture, whereas for cracked DENT specimens, the pre-existent cracks imposes high stress concentrations at the crack tip and the plastic deformation is limited to a more narrow plastic zone.

The experimental testing of the DENT specimen was conducted following the essential work of fracture methodology described in the work of Casellas et al. [11]. Both force-displacement data and DIC data were generated during tensile testing of DENT specimen which provided detailed information about the load level, displacement and work of fracture needed for the opening and propagation of the crack.

### 3.4. Numerical Modelling

LS-Dyna was used to perform numerical models of the triaxiality tensile testing, VDA 238-100 and DENT testing of the CP1000HD sheet metal. Discretization of the various sheet metal geometries was done with fully integrated 8-nodes solid hexahedral elements. The characteristic element length at the area of localisation and the failure of the specimens was L=0.1 mm, thus the same length as the DIC facet size used for failure strain measurement. By using the same characteristic element length as the DIC facet size, it was possible to avoid mesh size dependent failure behaviour. The hexahedral elements were uniformly arranged in the deformation and failure zone of the various specimens. All preparation of the FE-mesh was made using the ANSA pre-processor. The models used explicit time integration in order to efficiently handle the high non-linearity caused by material failure and element erosion. Termination times of the simulations were defined as sufficiently short to avoid overly costly computations, but long enough to keep dynamic effects to a minimum. Short termination times could only be used, as the CP1000HD material model deliberately did not include strain rate effects.

#### 3.4.1. Triaxiality Tensile Testing

Simulations of the triaxiality testing specimens in Figure 1 were performed initially to validate the failure and damage model. While the R3.75 specimen constituted the calibration case for the plastic hardening curve and damage model, the remaining three tensile test specimens served as validation cases. The tensile testing setup utilised a 50 mm long strain gauge placed centred at the specimen geometry. Assuming that the vast majority of the deformation occurs at the notches, only 50 mm adjacent to the centre of the specimen geometry was modelled, matching the placement of the strain gauge. An exception was made for the modelling of the shear specimen where the entire specimen geometry was used in order to capture the in-plane rotation of the localisation zone. A prescribed motion was applied to one of the specimen ends, while the other specimen end remained fixed. Figure 9 shows the R3.75 FE-model.

#### 3.4.2. VDA 238-100

Figure 10 shows the FE-model of the VDA 238-100 plate bending test. As stated in the VDA 238-100 standard described in Section 3.3.1, the model consists of a sample, two rollers and a punch with a 0.4 mm radius. All tools were considered rigid with no friction in the contact between sample and rollers. To reduce the computational cost, only a 1 mm strip of the sample was modelled with out-of-plane symmetry boundary conditions prohibiting out-of-plane motion and retaining the plane strain conditions of the model sample strip.

#### 3.4.3. Double Edge Notched Tensile (DENT) Test

The modelling of the DENT testing resembled the tensile testing models presented in Section 3.4.1. However, the DENT model consisted of a symmetry boundary condition along the specimen centreline with the purpose of lowering the computational cost. The modelling of the fatigue crack consisted of creating duplicate nodes along the fatigue crack, as done by Lindström et al. [41] and Jonsén et al. [42] during J-integral analysis in LS-Dyna. Figure 11 shows the DENT FE-model, where the duplicate nodes of the fatigue crack are highlighted with yellow circles.

## 4. Results

Figure 12, Figure 13, Figure 14, Figure 15 and Figure 16 show the simulation results for triaxiality tests, VDA 238-100 and notched and cracked DENT specimens. Together with the simulation results, corresponding experimental results are included for comparison.

Since the CP1000HD damage and failure model was calibrated for matching the force-displacement results from the R3.75 tensile testing, the correlating results between simulation and experiment in Figure 12a are expected. However, for the remaining tensile tests, a similar correlation is found, as shown in Figure 12b–d.

Figure 13 shows the simulation results for the VDA 238-100 model along with the four sets of experimental results. As seen in Figure 13, there is a clear correlation between the FE-model results and the experimental results both in terms of force-displacement response and for predicting the bending punch stroke until failure. Figure 14 shows the observed cracks from experiments compared to the crack initiation predicted by simulation, defined as deleted elements.

The comparison between experimental and simulation results for notched DENT specimens are shown in Figure 15. The comparison in Figure 15 shows that the modelling of the notched DENT specimen correlates well with the experimental results until the final part of the fracture process. The points of crack initiation are marked in Figure 15 and Figure 16, where (•) and (∘) identifies the crack initiation at the specimen surface from FE-models and experiments, respectively. In tensile testing of DENT specimens, it is well reported that the crack initiates at the centre of the specimen, where stress triaxialtiy is higher than the outer free-surfaces of the specimen, as stated by Frometa et al. [40]. This implies that crack initiation, in the specimen centre, is experimentally difficult to assess as the experimental DIC procedure could only capture data from the specimen surface. Thus, many works report crack initiation when it reaches the specimen’s surface. However, modelling can provide this crack initiation point and give relevant information for cracking prediction. The asterisk mark (∗) in Figure 15 and Figure 16 denotes the first point of crack initiation predicted by numerical models, occurring at the centre of the specimen. As stated, crack initiation in the notched DENT specimen centre was not possible to detect experimentally, which is why the validation of the numerically obtained crack initiation of the specimen centre could not be done. Comparing points of crack initiation at the specimen’s surface between numerical models and experiments show a decent correlation. Meanwhile, it is shown in Figure 16 that the numerical modelling of cracked DENT specimens could not accurately reproduce the force displacement results from experiments. The results show an over-prediction of displacement at surface crack initiation with 15–35%, where the largest deviation occurs for the shortest ligament lengths. Considering the poor correlation between the modelling and experimental results of cracked DENT specimens, it can be stated that the current damage and failure model could not accurately predict the fracture behaviour of specimens containing cracks. The results in Figure 16 show that inaccurate crack initiation from modelling produces overly high forces for the remaining displacement. This trend applies for the entire range of ligament lengths.

Comparing the equivalent strain field of the DENT ligament for both cracked and notched DENT specimen right before crack initiation is a supplementary indication of the modelling accuracy. Figure 17 shows the equivalent strain field for notched DENT specimen with a ligament length of 6 mm. Similarly, Figure 18 shows the equivalent strain field for notched DENT specimen with a ligament length of 5.087 mm. A similar set of images is presented in Figure 19 for notched DENT specimen with ligament length of 14 mm, and Figure 20 for cracked DENT specimen with ligament length of 13.213 mm.

## 5. Discussion

The good correlation between experiments and simulation for VDA 238-100 and notched DENT in Section 4 shows that the damage and failure model accurately predicts crack initiation where undamaged sheet steel of CP1000HD grade is subjected to tensile and bending deformation. The CP1000HD damage and failure model presented in this article is therefore useful for predicting failure through thinning and necking in, for instance, forming applications as drawing or deep drawing. From Figure 15, it is also apparent that the damage and failure model is capable of reproducing the fracture evolution process of the notched DENT specimen with decent accuracy, almost until the final fracture of the specimen.

However, the study of cracked DENT specimens in Figure 16 shows poor simulation accuracy in defining the crack initiation. The numerical models of the cracked DENT tests all experience over-predicted displacement for crack initiation compared to experiments, with an average of 28% over-prediction, which is why the following fracture evolution process will inherit inaccurate force levels. Numerical modelling of crack initiation and propagation generally imposes stringent requirements on the FE-discretisation. The crack-tip needs to be sufficiently resolved and structured in order to account for the crack-tip singularity, and the crack propagation path is inevitably controlled by the mesh, as stated by Anderson [22]. Neither the mesh size nor the mesh structure of the numerical model of the cracked DENT specimens were sufficient to reproduce the crack initiation. The combination of damage and failure modelling and FE-discretisation used in this article causes a plastic process before crack initiation and the crack-tip singularity is not accurately treated. This inaccuracy is accentuated when comparing the ligament strain field for cracked DENT specimens shown in Figure 18 and Figure 20, where a more extensive strain field is obtained at crack initiation in a simulation compared to the experimental results. On the contrary, Figure 17 and Figure 19 show good ligament strain field accuracy for notched DENT specimens. An even finer mesh than used in this article would cause crack initiation at lower displacements as the crack-tip elements would reach failure strain faster, and thus generate an improved correlation between the simulation and the test for the cracked DENT specimens. However, smaller element lengths (L<0.1 mm) would generate premature failure for modelling of the triaxiality tensile specimens as well as for the VDA 238-100 model since the damage and failure model is calibrated towards an element length of L=0.1 mm. Through mesh regularisation of the damage and failure model, a finer mesh would generate similar results as a coarser mesh, with the drawback that cracked DENT modelling results would again diverge from experiments. Calibration of the failure surface to fit the separate case of cracked DENT specimens would solve the over-predicted displacement at crack initiation for this particular problem, but would mean that the generality of the material characterisation method used in this article is lost. Failure surface calibration to specific problems would also infer inconvenience when using the CP1000HD damage and failure model on geometries where cracks emerge. This case is, for instance, found in the modelling of shear cutting, where the shearing process is followed by the opening of cracks preceding the fracture process, as shown by Dalloz et al. [43], Wang and Wierzbicki [44] and Zhang et al. [45] in interrupted punching tests.

## 6. Conclusions

In this study, the capabilities of predicting crack initiation and propagation using a strain-driven damage and failure model were evaluated through correlation with experiments consisting of both cracked and un-cracked specimens. The following conclusions were drawn:This work shows that strain-driven ductile failure modelling of crack-tip analysis cannot be used to predict the crack initiation of cracked high strength sheet metal. The results show an over-prediction of roughly 28%, which consequently causes the following crack propagation to be invalid as well. This holds even if the failure model accurately describes the large scale failure of undamaged material. As the numerical modelling presented in this article fails to describe the fracture initiation at the crack tip, it sets the basis for further development of fracture models to overcome the identified limitation.The experimental and numerical results presented in this article identify the need for a damage and failure model that extends strain-driven ductile failure models such as GISSMO with the capabilities of treat cracked materials with a triaxiality dependent ductile fracture. This extension enables the refinement of fracture mechanical testing procedures, such as the EWF methodology, if the numerical modelling can provide insight into crack nucleation occurring at positions hidden from the DIC camera.Furthermore, an extended strain-driven ductile failure model is required to enhance the accuracy when modelling sheet manufacturing processes involving crack propagation phenomena. This is the case of many forming processes, where the sheet is fractured and separated, such as trimming, blanking, punching, and so forth. When high strength materials, such as AHSS or high strength Al alloys, are manufactured, the part quality is largely affected by the crack propagation resistance exhibited by the material. Such behaviour implies that material models should accurately account for the crack initiation and propagation.Additionally, improved capabilities of modelling crack initiation and propagation will consequently aid in rationalising cracking behaviour in metallic sheets. It will definitively help to predict and minimize the edge cracking phenomenon that originates from sheared edges and is a great concern for part manufacturers when using high strength materials.

## Figures and Tables

**Figure 1 materials-14-07821-f001:**
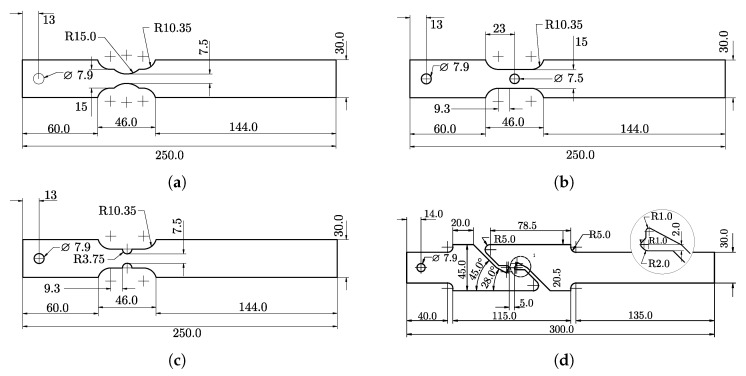
Tensile test specimens for triaxiality testing. (**a**) R15. (**b**) Hole. (**c**) R3.75. (**d**) Shear.

**Figure 2 materials-14-07821-f002:**
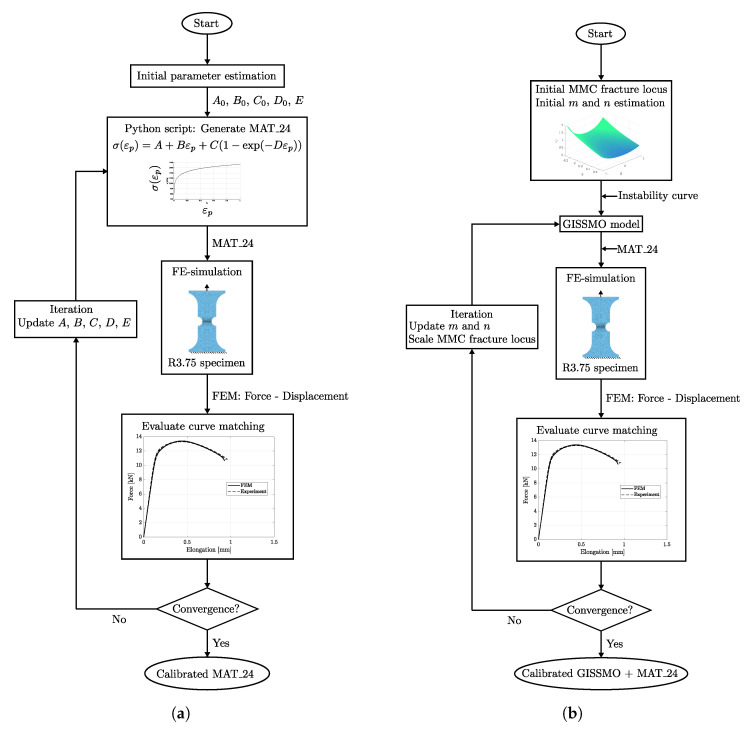
Calibration flowchart for CP1000HD plastic hardening curve and damage and failure model. (**a**) Plastic hardening model. (**b**) Damage and failure model.

**Figure 3 materials-14-07821-f003:**
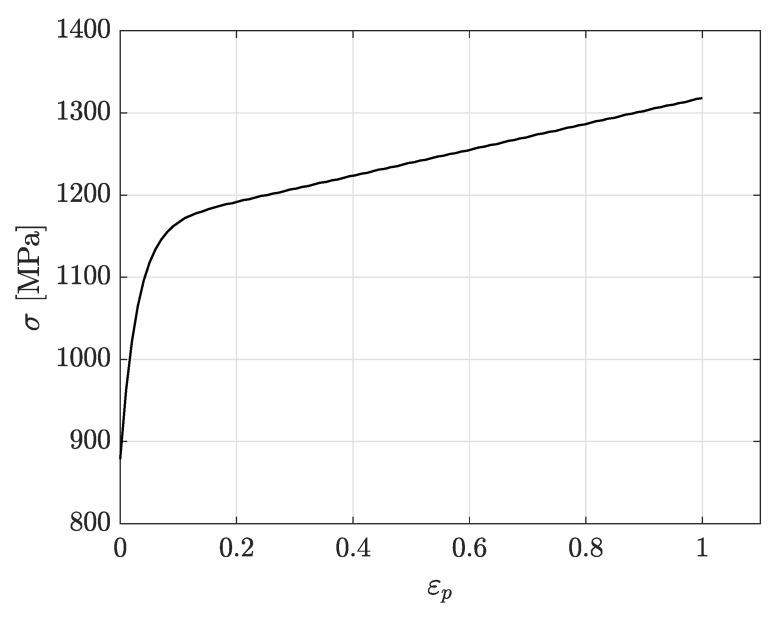
True plastic strain–true stress for CP1000HD $90^{\circ }$ to the rolling direction.

**Figure 4 materials-14-07821-f004:**
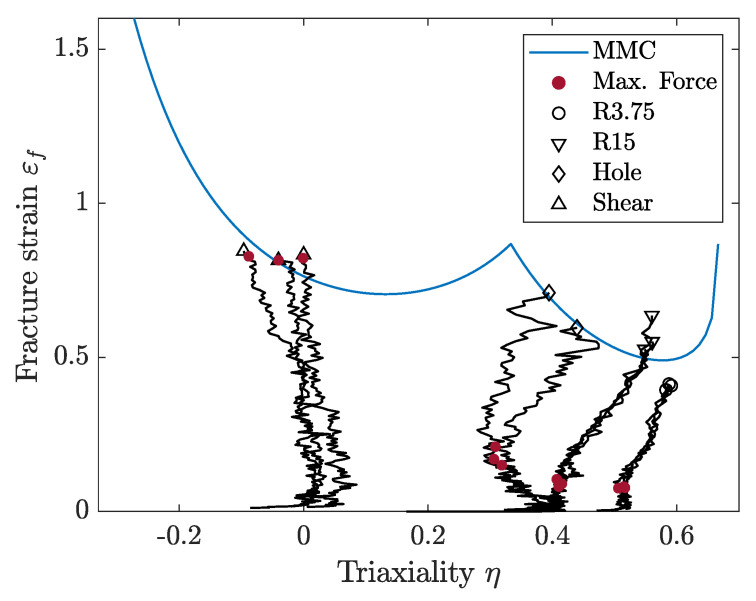
Modified Mohr–Coulomb failure curve for plane stress assumption.

**Figure 5 materials-14-07821-f005:**
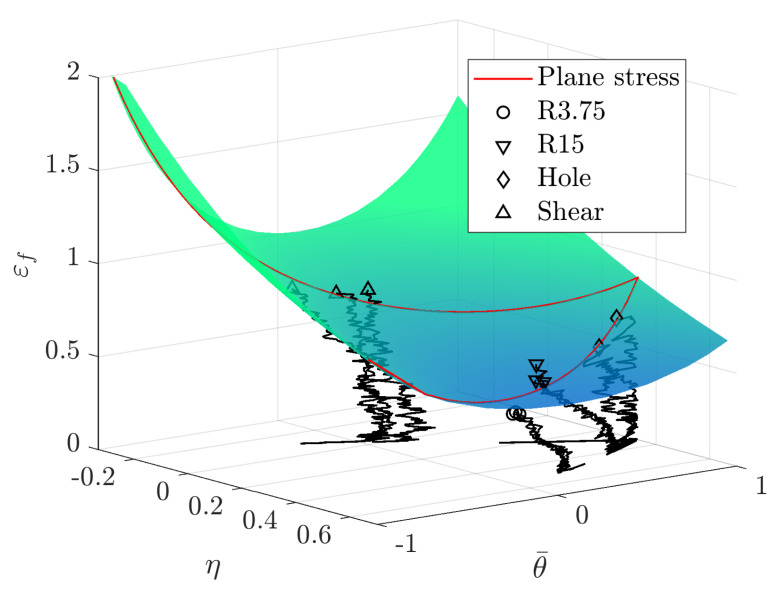
Modified Mohr–Coulomb failure surface.

**Figure 6 materials-14-07821-f006:**
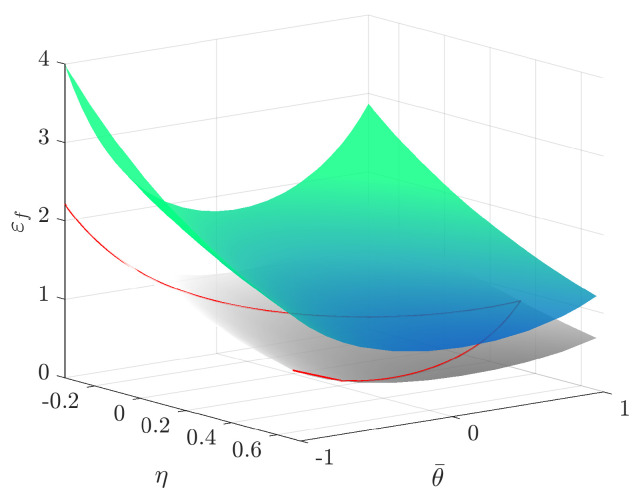
Original MMC surface (grey) and calibrated MMC surface (color).

**Figure 7 materials-14-07821-f007:**
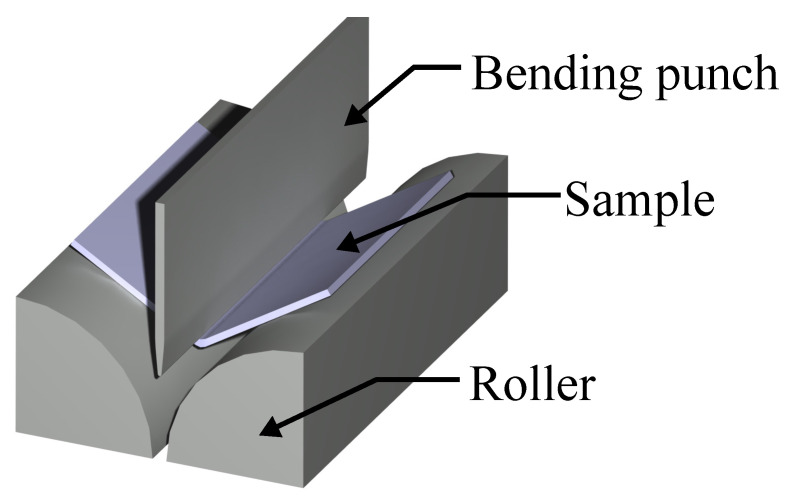
VDA 238-100 experiment setup.

**Figure 8 materials-14-07821-f008:**
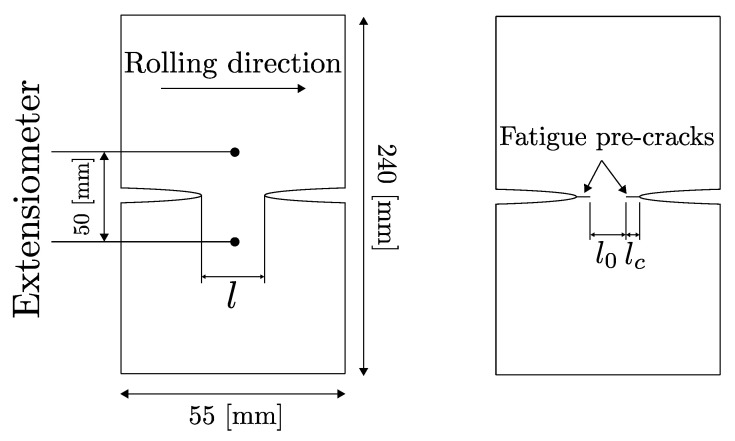
Notched and cracked DENT specimen geometry. Sheet rolling direction is indicated.

**Figure 9 materials-14-07821-f009:**
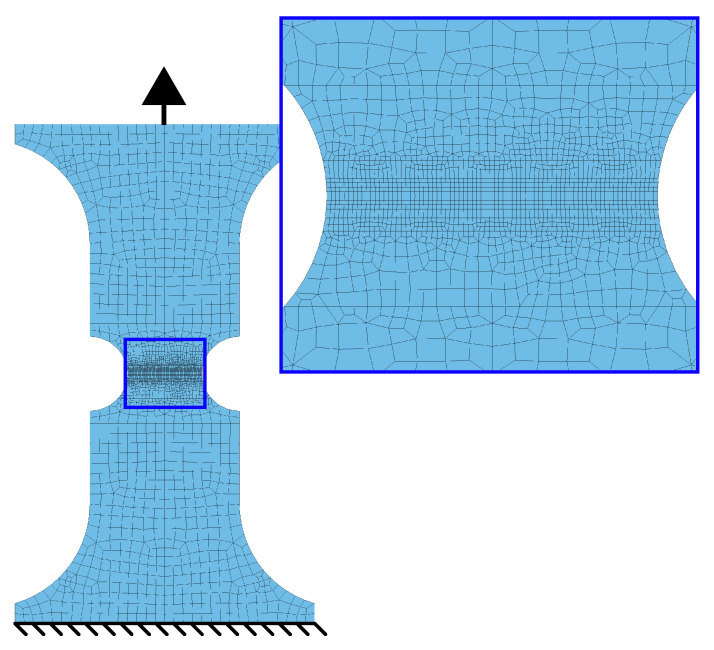
FE-model of R3.75 specimen.

**Figure 10 materials-14-07821-f010:**
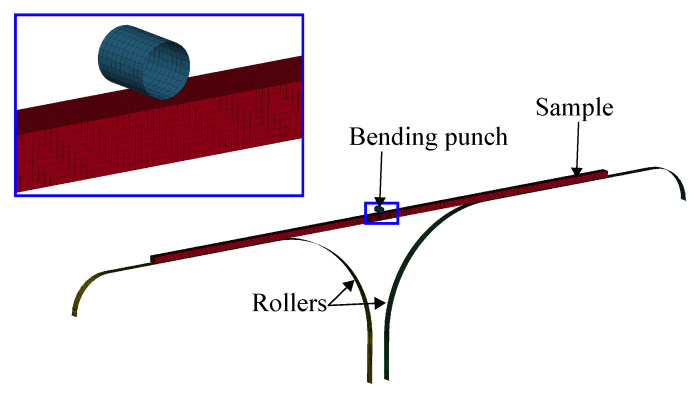
FE-model of VDA 238-100 bending test.

**Figure 11 materials-14-07821-f011:**
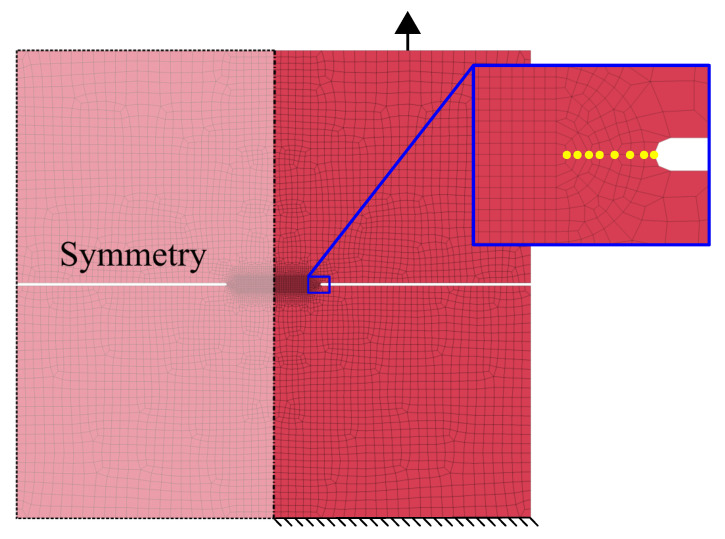
FE-model of DENT test with fatigue crack interpreted with duplicate nodes.

**Figure 12 materials-14-07821-f012:**
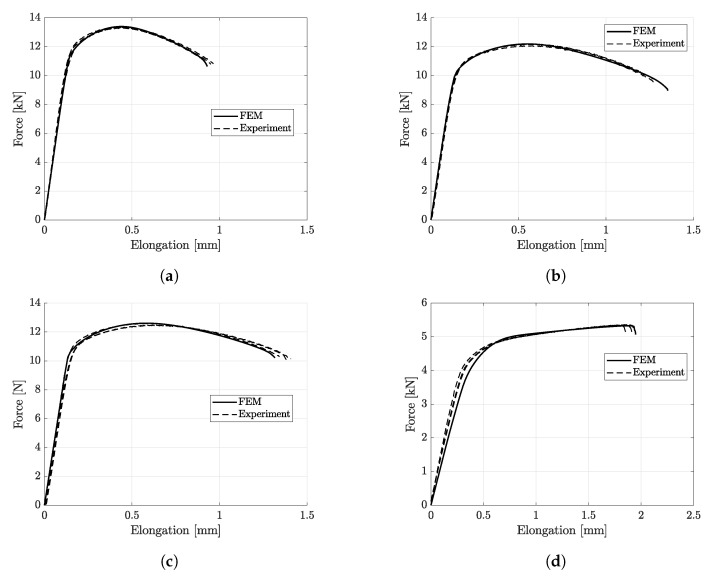
Experimental and simulation results of triaxiality testing. (**a**) R3.75. (**b**) Hole. (**c**) R15. (**d**) Shear.

**Figure 13 materials-14-07821-f013:**
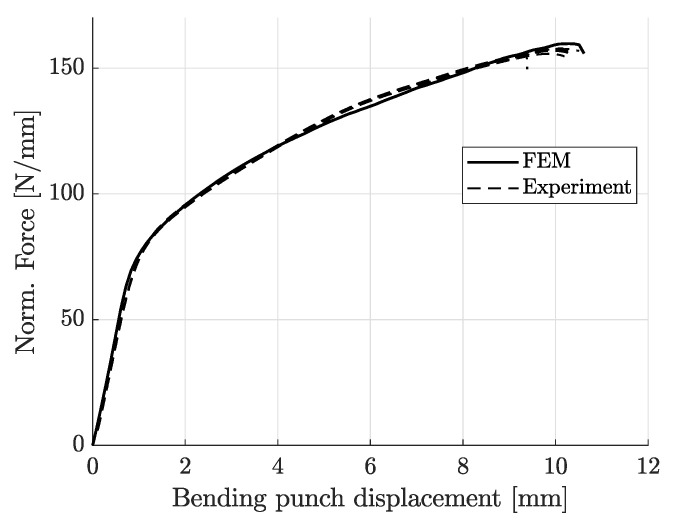
Experimental and simulation results of VDA 238-100 bending test.

**Figure 14 materials-14-07821-f014:**
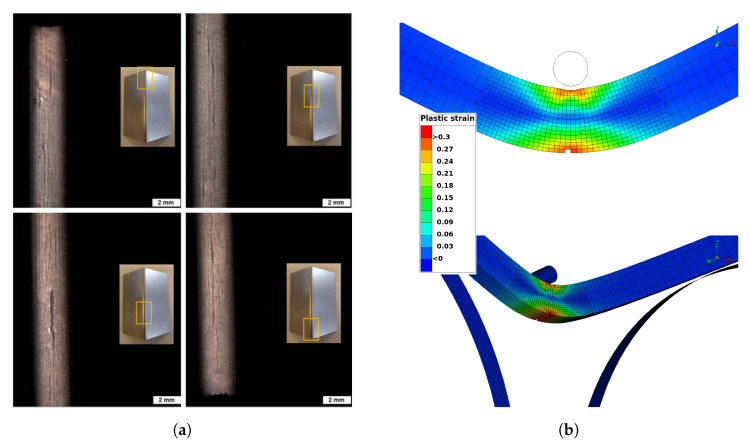
Observed cracks and comparison of crack position between experiments and simulations of VDA 238-100 bending test. Courtesy of Faurecia Autositze GmbH. (**a**) Experimental results. (**b**) Simulation results.

**Figure 15 materials-14-07821-f015:**
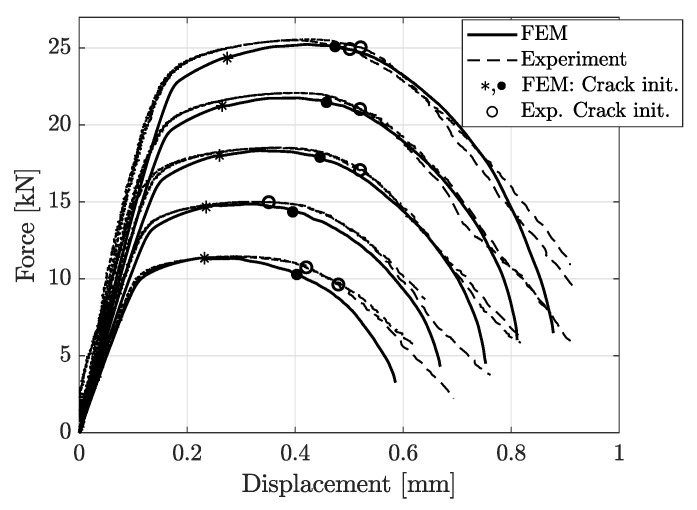
Experimental and simulation results of notched DENT test, ranging from $l_{6\mathrm{mm}}$ (lowest peak force) to $l_{14\mathrm{mm}}$ (highest peak force). Onset of fracture at the specimen surface is marked with (•) and (∘), while crack initiation inside the specimen is marked with (∗).

**Figure 16 materials-14-07821-f016:**
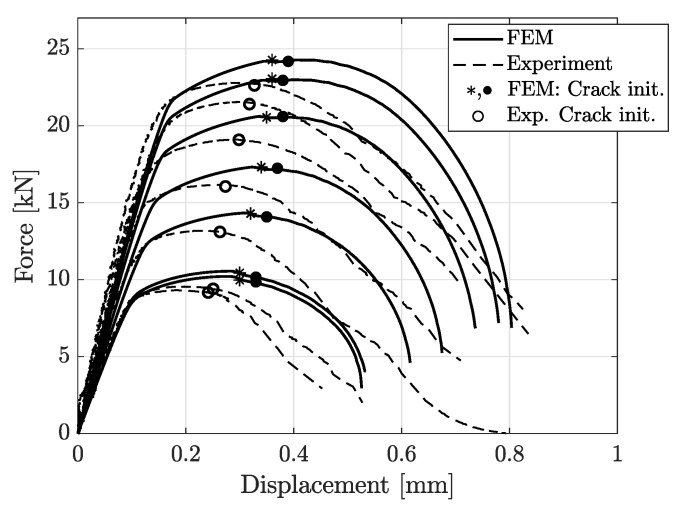
Experimental and simulation results of cracked DENT test, ranging from $l_{8\mathrm{mm}}$ (lowest peak force) to $l_{16\mathrm{mm}}$ (highest peak force). Onset of fracture at the specimen surface is marked with (•) and (∘), while crack initiation inside the specimen is marked with (*).

**Figure 17 materials-14-07821-f017:**
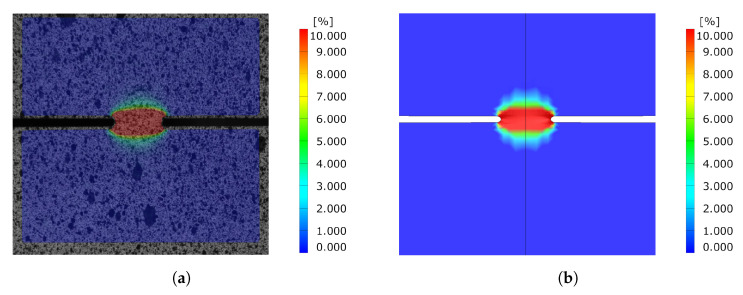
Comparison of equivalent strain field between experiment and simulation before crack initiation for Notched DENT specimen with l=6 [mm]. (**a**) Experimental results. (**b**) Simulation results.

**Figure 18 materials-14-07821-f018:**
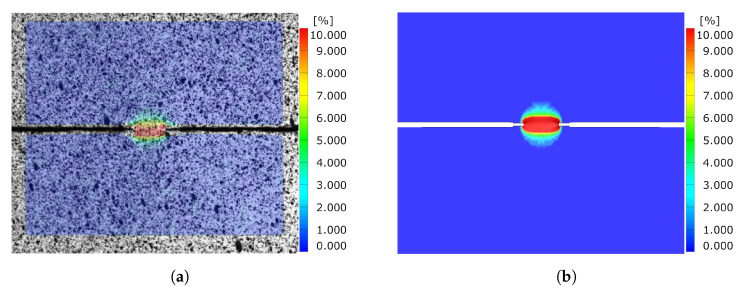
Comparison of equivalent strain field between experiment and simulation before crack initiation for cracked DENT specimen with l=8 [mm] and $l_{0}=5.087$ [mm]. (**a**) Experimental results. (**b**) Simulation results.

**Figure 19 materials-14-07821-f019:**
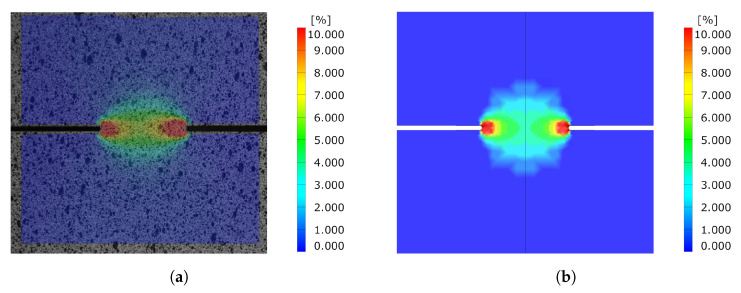
Comparison of equivalent strain field between experiment and simulation before crack initiation for Notched DENT specimen with l=14 [mm]. (**a**) Experimental results. (**b**) Simulation results.

**Figure 20 materials-14-07821-f020:**
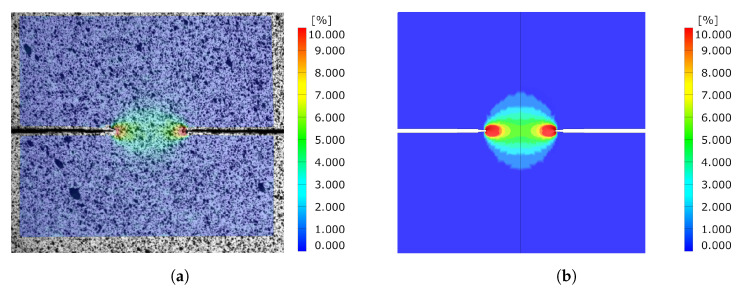
Comparison of equivalent strain field between experiment and simulation before crack initiation for cracked DENT specimen with l=16 [mm] and $l_{0}=13.213$ [mm]. (**a**) Experimental results. (**b**) Simulation results.

**Table 1 materials-14-07821-t001:** Plastic hardening material constants of the Stiebler model.

Parameter	*A* [MPa]	*B* [MPa]	*C* [MPa]	*D*
Value	878.6	157.9	281.8	34.1

**Table 2 materials-14-07821-t002:** Tensile properties at different rolling directions for CP1000HD (thickness of 1.5 mm), yield strength (Rp02), tensile strength (Rm), uniform elongation (Ag), total elongation (A80), n-value at uniform elongation (nAg) and r-value (*r*).

Test Direction	Rp02 [MPa]	Rm [MPa]	*Ag* [%]	*A*80 [%]	*nAg*	*r*
$0^{\circ }$	893	1052	7.3	11.1	0.071	0.91
$45^{\circ }$	905	1052	7.0	10.6	0.068	1.02
$90^{\circ }$	909	1062	7.1	10.7	0.068	0.96

**Table 3 materials-14-07821-t003:** MMC parameters for CP1000HD.

Parameter	$C_{1}$	$C_{2}$	$C_{3}$	$C_{4}$	$C_{5}$
**Value**	0.0992	1.7814	1.0247	1.0479	0.2127

**Table 4 materials-14-07821-t004:** Calibrated damage and failure model parameters.

Scale $\varepsilon_{f}$	*n*	*m*
1.8	4.0	3.0

**Table 5 materials-14-07821-t005:** DENT specimen ligament lengths.

	Notched	Cracked
**Test nr.**	l **[mm]**	l **[mm]**
1	6	5.273
2	8	5.087
3	10	7.387
4	12	9.100
5	14	11.042
6		12.458
7		13.213

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
