# Peer review of "Stating Failure Modelling Limitations of High Strength Sheets: Implications to Sheet Metal Forming"

_materials, 2021, doi:10.3390/ma14247821_

Round 1

Reviewer 1 Report

This paper discusses the fracture modelling accuracy of strain-driven ductile fracture models when introducing damage of high strength sheet steel. It sets the basis for further development of fracture models to overcome the identified limitation. The work is interesting and meaningful. This paper should be accepted, but there are some issues needed to be addressed:

  1. In tensile tests, how to determine the stress triaxiality of the different notched specimen geometries?
  2. In Fig.4 and Fig.5, more fluctuations occurred in the tensile tests for different stress triaxiality specimens. Why the curves are fluctuating?
  3. In this paper, the author obtained the Modifed Mohr-Coulomb (MMC) failure surface of the material. However, the in-depth analysis focused on the GISSMO failure- and damage model. Why the authors introduced the MMC?
  4. Quantified results cannot be found in the abstract and conclusions.
  5. In the conclusions, we cannot find the encouraging contents. It is suggested that the author reorganize the conclusions.

Reviewer 2 Report

Based on the Generalised Incremental Stress State Dependent Damage Model, this work try to make a extension of the the GISSMO failure- and damage model,  to better improving of fracture mechanical testing procedures and modelling of high strength sheet metal manufacturing.

Generally, I do think that the manuscript is well prepared.
The topic of damage model is an endless story in metal forming erea, especailly in sheet metal forming.
It is still a very hot research topic and there are lots of publications in recent years.

Of course, the Generalised Incremental Stress State Dependent Damage Model is not new damage model.
However, the thorough analysis a group of experimentla tests and simualtions in this work could help to better understanding this useful model.

In the manuscript the Fig.2 and Fig.6 is quite simliar, this is a point needed to be improved.

Reviewer 3 Report

In this work, failure modelling limitations of high strength sheets has been stated. This paper is thorough, original and technically relevant. This paper is also easy to follow, has representable figures, and references cited appear to be in proper order. However, some issues are mentioned here which need to explain by the authors.

  1. The introduction or GISSMO damage model section should incorporate some literature where the recent advancement on the various damage models should be discussed. Moreover, the recent advancement on the implementation of the anisotropy of the sheet metal on the damage model also needs to be discussed. There are several works where the anisotropy of the sheet metal has been directly incorporated into the ductile damage model. This recent trend will make the manuscript more technically enriched. Authors must bring out these points so that the work also looks more relevant comparing with the current trends.

From a quick glance I can refer some recent work the above mentioned topic which may help authors. Park, N et al. 2017. International Journal of Plasticity, 96, pp.1-35.,

Basak, S. et al. 2020. International Journal of Mechanical Sciences, 170, p.105346.,

Soeiro, J.M.C., et al., 2015. Journal of Materials Processing Technology, 217, pp.184-192 etc.

  1. Authors can list down the sheet material experimental properties including YS, UTS, elongation, r-values along various anisotropic directions in this manuscript. What is the thickness of the sheet metal?
  2. How the design of the tensile specimen for triaxiality testing has been fixed? Has it taken from any other literature? Or these are designed for this work. Please justify.
  3. How the authors have done non-uniform solid meshing scheme of the fracture specimens? Which preprocessor has been used? How the optimum element length in the highly deforming region/failure region has been determined? Is any mesh optimization method has been used?
  4. From figure 4 it can be observed that the triaxiality state at the fracture point was negative. Hence, is it worthy to use this kind of geometry of the in-plane shear fracture specimen to calibrate the MMC model whose triaxiality state at the fracture point is not actually ZERO?
  5. All the fracture specimens have been prepared along which anisotropic directions? Does the material anisotropy has been considered during fracture locus calibration? If no, why? Please justify.

Reviewer 4 Report

The manuscript reads well and highlights an important shortcoming of ductile fracture models for simulating fracture in initially cracked structures. The reviewer noted that there are some points in fracture model calibration, which were not described in good details. These are as follows:

  1. In Fig. 2, the FE model of the specimen used for hardening curve calibration is not described well. A separate, dedicated sub-section is needed for FE modeling of all specimens, and discussions should be included on how well the FE results match the test results when using the calibrated hardening curve (this should be before Section 4, where the results for VDA 238 and DENT tests are presented).
  2.  The authors used DIC technique to extract loading paths from the tests in terms of equivalent plastic strain, stress triaxiality and Lode angle parameter variations. DIC can measure only surface strains, whereas in the test specimens R15, Hole and R3.75 fracture is expected to initiate in the mid-surface (not on the surface) after localized necking (thinning). The authors should provide more explanations on the experimental measurements, how they derived stress triaxiality and Lode angle parameter, and where exactly fracture initiated in the test specimens.
  3.  In Figure 1, the given stress triaxiality values are not meaningful as the loading paths are non-proportional (stress triaxiality varies with straining).
  4.  Lines 193-202. I do not understand why plane stress assumption made for the calibration of MMC model.

Minor editorial comments:

  1. In Table 1, the units of the constants A, B, and C should be stated (MPa).
  2. At several sentences, the word "through" is typed as "trough". Please scrutinize the manuscript for typos before re-submission.

Round 2

Reviewer 4 Report

Thank you for your responses. All my concerns were adequately addressed.